# *vapD* Mutation Shows Impairment in the Persistence of *Helicobacter pylori* Within AGS Cells

**DOI:** 10.3390/microorganisms13081952

**Published:** 2025-08-21

**Authors:** Rosario Morales-Espinosa, Gabriela Delgado, Carlos A. Santiago, Alejandro Flores-Alanis, Rafael Diaz-Mendez, Alberto Gonzalez-Pedraza, José L. Méndez, Alejandro Cravioto

**Affiliations:** 1Laboratorio de Genómica Bacteriana, Departamento de Microbiología y Parasitología, Facultad de Medicina, Universidad Nacional Autónoma de México, Mexico City 04510, Mexico; delgados@unam.mx (G.D.); santiagoolivares@unam.mx (C.A.S.); bioalejandrofa@gmail.com (A.F.-A.); albemari@unam.mx (A.G.-P.); joselu@unam.mx (J.L.M.); dracravioto@hotmail.com (A.C.); 2Centro de Ciencias Genomicas, Universidad Nacional Autónoma de México, Mexico City 04510, Mexico; rdiaz@ccg.unam.mx

**Keywords:** VapD protein, *Helicobacter pylori*, *H. pylori* 26695 *vapD* mutant

## Abstract

The *Helicobacter pylori vapD* gene is transcribed and expressed when the bacteria are within the gastric cell. In this current study, we investigated how *vapD* knockout affects the survival of *H. pylori* inside human gastric adenocarcinoma cells. We constructed an *H. pylori* 26695 *vapD* (Hp Δ*vapD*) mutant strain. *H. pylori* 26695 wt and Hp Δ*vapD* strains were grown in synthetic media and were co-cultured with AGS cells. From the start, the growth curve, total protein concentration and colony-forming units (CFUs) of each strain were measured. From each co-culture, CFUs and total RNA were obtained, and transcript levels of GAPDH, *vapD*, *vacA*, *ureA*, and 16s Hp were measured by qRT-PCR. Hp Δ*vapD* did not affect the growth rate of the strain in synthetic media, showing that the *vapD* gene is not necessary when the bacteria grow outside eukaryote cells. However, in the intracellular environment, the number of CFUs recovered from the Hp Δ*vapD* strain from AGS cells decreased after 36 h. Transcription levels of the *vacA* gene from the Hp Δ*vapD* strain were 10,000-fold lower than those of *H. pylori* wt, to the point of being undetectable. The results suggest that the *vapD* gene contributed to maintaining *H. pylori* inside gastric cells.

## 1. Introduction

*Helicobacter pylori* is a Gram-negative bacterium that colonizes the human gastric mucosa. There are different *H. pylori* genotypes, which have important genetic differences and clinical implications, with some being more prevalent in specific geographic regions [1]. *H. pylori* is responsible for producing atrophic gastritis and peptic ulcers, and it is an important contributing factor to the development of gastric cancer [2]. For practical purposes, the strains of *H. pylori* can be divided into two main genotypes: genotype I and genotype II [3]. While *H. pylori* genotype II stains lack virulence genes and colonize the epithelial cells of the stomach of asymptomatic patients who can carry the bacterium for decades without any clinic outcome, *H. pylori* genotype I strains are associated with the development of various severe gastric pathologies, since these strains possess genes that encode for the virulence factor, such as the *vacA* gene that encodes for vacuolating cytotoxin A; the cag pathogenicity island [*cag*-PAI (cytotoxin-associated genes pathogenicity island)], which encodes a type IV secretion system (T4SS); the *cagA* gene that encodes for the CagA oncoprotein, which is translocated within the epithelial cell through T4SS, altering intracellular signal transduction pathways and probably contributing to oncogenesis [2]; and different adhesins.

Recently, our research group studied the *vapD* (virulence-associated protein) gene of *H. pylori* (strain 26695) in co-cultures with human gastric adenocarcinoma (AGS) cells using RT-PCR, and we found that the *vapD* gene was transcribed at high levels when the bacteria interact with the eukaryote cell. In this same study, we showed that *vapD* was transcribed in gastric biopsy samples (antrum and corpus) from patients with severe gastric diseases such as atrophic chronic gastritis, follicular gastritis, peptic ulcers, and gastric cancer [4]. Another recent study demonstrated that the *vapD* gene is not only transcribed but the VapD protein is also expressed in the intracellular environment of AGS cells, maintaining its expression *(manuscript in preparation*) over an indefinite time. The VapD protein mechanism of action is not known; however, it has been attributed to endoribonuclease activity [5].

The *vapD* gene has been described in other microorganisms, such as *Rhodococcus equi*, a Gram-positive pathogen associated with life-threatening bronchopneumonia in foals; *Haemophilus influenzae*, a Gram-negative microorganism associated with otitis media and respiratory disease in humans; *Neisseria gonorrhoeae*, a Gram-negative bacterium associated with gonorrhea in humans; and *Dichelobacter nodosus*, a Gram-negative bacterium responsible of ovine foot rot, all of which are facultative intracellular microorganisms that have developed strategies to remain inside eukaryotic cells [6,7,8,9]. In different microorganisms, the *vapD* gene is located on the chromosome as a single gene or as part of a toxin–antitoxin (TA) system or on a plasmid as part of a group of genes of the *vap* family (*vapA-VapH*). For example, in *H. influenzae,* the *vapD* gene is part of the *vapXD* (TA) system, which is involved in the stress-induced post-transcriptional regulation of gene expression via the mechanism of mRNA cleavage to maintain the survival and virulence of NTHi under stress conditions such as starvation, as well as heat, osmotic and free radical-induced stress. In *R. equi*, the VapA and VapD proteins are encoded by the *vapA* and *vapD* genes present in a virulence plasmid that are overexpressed under acidic conditions. The presence of other Vap proteins that participate in the virulence of the microorganism has been documented, such as VapK and B (produced by plasmids often found in porcine species) or VapN (produced by plasmids often isolated in bovine and human samples), all of them key for the intracellular survival of *R. equi* in the host macrophages [10]. The virulence or intracellular capacity of these microorganisms is due to the presence of the *vapD* gene, since when it is mutated in *R. equi* and *H. influenzae*, the microorganisms cannot survive in macrophages [6,8] or in respiratory epithelial cells, respectively. It should be noted that in some strains of *N. gonorrhoeae*, *vapD* is present in a cryptic plasmid (pCryp) that is part of the putative *vapDX* system and in the chromosome, located within a putative VirB T4SS locus, although *N. gonorrhoeae* usually has two *vapD* genes, one of them in a pConj and the other associated with a pCrip; in both cases, the *vapD* genes have homology with the CRISPR/Cas-associated Cas2 protein, which forms a complex with Cas1, which provides immunity against invading mobile genetic elements [11]. In the case of *D. nodosus* and *Riemerella anatipestifer*, the *vapD* gene is present in a plasmid, and when these microorganisms are released from the plasmid, they lose their virulence [9,12] and are unable to cause disease. In this current study, we wanted to know if the survival or persistence of *H. pylori* (strain 26695) within the AGS cells is affected by knocking out the *vapD* gene. Our results show that the *vapD* gene mutant of *H. pylori* prevented the survival of *H. pylori* within the cell, concluding that *vapD* is playing an important role in maintaining the bacterium in the intracellular environment and that it probably contributes to the chronicity of the infection.

## 2. Materials and Methods

### 2.1. Bacterial and Cell Culture

*H. pylori* 26695 (ATCC 700392) was grown in Brucella agar plates with 5% sheep blood and 5% fetal bovine serum (Biowest) under humidified and microaerophilic conditions (5% O_2_, 8% CO_2_ and 85% N_2_) at 37 °C for 48 h.

AGS cells (ATCC 1739) were grown in Dulbecco’s Modified Eagle Medium (DMEM) (Gibco) supplemented with 10% fetal bovine serum (Biowest) and incubated under humidified atmosphere of 5% CO_2_ at 37 °C.

### 2.2. Molecular Cloning of vapD and Chloramphenicol Resistance Cassette

#### 2.2.1. Molecular Cloning of *vapD* (HP0315) Gene

*H. pylori* 26695 chromosomal DNA was used for the amplification of the *vapD* (HP0315) gene by PCR using Platinum HF DNA polymerase (Invitrogen, Carlsbad, CA, USA) with primers 1F and 1R [13] (Table 1). The 285 bp amplified product was cloned into the pCR 2.1 vector (Invitrogen) to generate the p*vapD* plasmid, where the *vapD* gene was flanked by *Eco*RI sites (Appendix A). The construct was transformed into *E. coli* DH5α competent cells, and the correct insertion of the *vapD* gene locus was confirmed by PCR analysis.

#### 2.2.2. Molecular Cloning of Chloramphenicol Resistance Cassette (*Cm^r^*)

The chloramphenicol resistance cassette (*Cm^r^*) was obtained from a chloramphenicol-resistant strain of *E. coli* O42 by PCR using the primer sets *cat*SpeIF and *cat*SpeIR (Table 1) and Platinum HF DNA polymerase (Invitrogen). The PCR product (900 bp) with its *Spe*I flanking sequence was cloned into pCR 2.1 vector (Invitrogen); the resulting p*cat*Spe plasmid was transformed into *E. coli* DH5α competent cells (Appendix A). The correct sequence of *Cm^r^ cassette* was confirmed by sequencing using universal primers M13F and M13R.

### 2.3. Construction of vapD (HP0315) Knockout Mutant

#### 2.3.1. Site-Directed Mutagenesis in *vapD* Gene

To generate the *vapD* gene disruption mutant, the p*vapD* plasmid was subjected to site-directed mutagenesis to create a unique *Spe*I restriction site (position 117 bp of *vapD*) using the primer sets *vapD*SpeF and *vapD*SpeR (Table 1). To determine if the *Spe*I restriction site was created, the resulting pPC*vap*Spe plasmid was transformed into *E. coli* DH5α. The correct sequence and direction were confirmed by sequencing using the universal primers M13F and M13R.

The p*vap*Spe plasmid was then digested with *EcoR*I, and the DNA fragment containing the unique restriction site *Spe*I was subcloned into the pK18mobsacB vector (ATCC 87097) to obtain the pk18-*vap*Spe plasmid. The chloramphenicol resistance cassette (*Cm^r^)* obtained from the p*cat*Spe plasmid was then inserted into the *Spe*I site of the pk18-*vap*Spe plasmid.

To disrupt the *vapD* gene and create the *vapD* knockout construction, a chloramphenicol resistance cassette (*Cm^r^)* was inserted into the unique *Spe*I site of the pk18-*vap*Spe plasmid, giving rise to the pk18-*vapD*::*Cm^r^* plasmid. This plasmid was transformed into *E. coli* DH5α. The successful insertion of the resistance cassette was confirmed by PCR, and the resistant colonies were selected on Brucella agar plates of sheep blood containing 30 µg/mL of chloramphenicol (Figure 1).

#### 2.3.2. Introduction of Homologous Flanking Regions by Overlap Extension PCR

To increase the probability that homologous recombination would take place between the knockout construction and the *H. pylori* 26695 wt strain, we introduced the 5′and 3′ flanking regions of *vapD* wt into the knockout *vapD*::*Cm^r^* construction, with approximately 1000 bp in each region.

To obtain the specific homologous flanking regions of the wild-type *vapD* gene, the chromosomal DNA from *H. pylori* 26695 wild-type was amplified by PCR using primers 3.5F and 3.5R (Table 1) to obtain a sequence of 3.5 Kb (PCR1). Six chimeric primers were designed with additional 5′ sequences to introduce homologous ends into the fragments to be fused (Table 1). The Lasergene 7 software was used to construct the primer sequences. Six PCR products with distinct overlapping ends were achieved (Figure 1). Additionally, the *vapD*::*Cm^r^* construction was amplified with 1F and 1R primers [13] (Table 1). The amplifications were carried out using Accuprime Pfx DNA polymerase (Invitrogen). The conditions for the reactions are shown in Appendix A. All PCR products were analyzed by agarose gel electrophoresis and purified by Nucleospin gel and a PCR clean-up kit (Macherey–Nagel).

To ligate PCR products of the homologous DNA fragments upstream (PCR3) and downstream (PCR4) of the *vapD* wt gene to the *vapD*::*Cm^r^* construct (PCR2), we used an overlap extension PCR method (Appendix A). In this method, 1 μL of PCR3 upstream DNA fragment and 1 μL of PCR2 (*vapD***::***Cm^r^*) were mixed along with Accuprime Pfx DNA polymerase and cycled without primers. This was followed by the fused DNA amplification phase for which a new PCR reaction was carried out with appropriate primer sets (5F and 6R), 2 μL of the overlap extension product (acted as template) and Accuprime *Taq* DNA polymerase High Fidelity (Invitrogen). The final PCR product of ~2.9 Kb (FUS1) was analyzed by agarose gel electrophoresis, cloned into a pCR 2.1 vector (Invitrogen) and transformed into *E. coli* DH5α.

Subsequently, the PCR4 fragment was joined downstream to the 3′ region of the *vapD*::*Cm*^r^ construction (PCR2) as previously described. Table 1 shows the primer set 7F/7R used to amplify PCR4, as well as the primers *cat*SpeIF/7*sal*R used in the fused DNA amplification phase. The PCR product obtained from DNA fusion (FUS4) was of 2.6 Kb and analyzed by agarose gel electrophoresis, purified by Nucleospin gel and a PCR clean-up kit (Macherey–Nagel), cloned into pCR 2.1 vector (Invitrogen) and transformed into *E. coli* DH5α.

Finally, the DNA FUS1 and FUS4 fragments were combined. As mentioned above, 1 μL of FUS1 and 1 μL of FUS4 were mixed with Accuprime Pfx DNA polymerase (Invitrogen) and cycled without primers (Appendix A). To recover the product from this final fusion, 1.5 μL of the overlap extension product (template) was amplified by PCR using the FUS32F/ FUS32R primer sets and Accuprime *Taq* DNA polymerase High Fidelity (Invitrogen) Appendix A. The final PCR product of ~3.6 Kb (FUS32) was analyzed by agarose gel electrophoresis, purified by Nucleospin gel and a PCR clean-up kit (Macherey–Nagel), cloned into pCR 2.1 (Invitrogen) and pK18-mobsacB (ATCC 87097) vectors, and transformed into *E. coli* DH5α. The correct sequence and direction of the pCRFUS32 plasmid were confirmed by sequencing, using the universal primers M13F/M13R and the specific primer sets FUS32 R5′F/FUS32 R5′R, FUS 32 RMF/FUS32 RMR and FUS32 R3′F/FUS32 R3′R (Table 1, Appendix A).

### 2.4. Electrotransformation and Homologous Recombination

#### 2.4.1. Preparation of Recipient *H. pylori* Cells

*H. pylori* 26695 wt strain was inoculated into ten Brucella agar plates with 5% sheep blood and supplemented with 5% fetal bovine serum. The plates were incubated under humidified and micro-aerophilic conditions (5% O_2_, 8% CO_2_ and 85% N_2_) at 37 °C for 36 h. After incubation, cells were harvested and suspended in 10 mL sterile water. The cell suspension was transferred to 50 mL conic tube and cold sterile water was added until a volume of 40 mL was achieved. The tube was then centrifugated at 3500 rpm at 4 °C for 10 min. The pellet was resuspended in 20 mL of cold 10% glycerol before being centrifugated as before, and resuspended in 14 mL of cold electroporation buffer (15% glycerol and 10% sucrose). Then the cells were centrifuged at 3500 rpm at 4 °C for 10 min, before removing the supernatant and resuspending it in 1 mL of cold electroporation buffer. The cells were aliquoted at 200 μL vol and prepared for electroporation. For long-term storage, competent cells were frozen in dry ice and stored at −80 °C until needed.

#### 2.4.2. Transformation and Recombination

To perform the transformation, 1 mL of the FUS32 construct was amplified by PCR. The PCR reaction was carried out using the specific primer set FUS32 F/R, the Accuprime Taq DNA polymerase High Fidelity (Invitrogen) and the pCRFUS32 plasmid as a template. The PCR product (1 mL) was purified with Nucleospin gel and a PCR clean-up kit (Macherey–Nagel), and the sample was eluted with 500 μL of ultrapure water.

A tube with frozen competent *H. pylori* 26695 cells was thawed on ice for 15 min. The 500 μL of purified FUS32 product was added and mixed with 200 μL of the competent cells’ suspension. The sample was chilled on ice for 5 min. Then, the bacterial suspension and FUS32 mixture was added very slowly to the bottom of a prechilled (0.2 cm gap) electroporation cuvette. The electroporation was performed with the Bio-Rad gene pulser, and it was set up at 2.5 kV, 25 μF capacitor and a resistance of 600 Ω in parallel, and the sample was subjected to single-pulse electroporation using a time constant of 12.5 ms.

After the pulse, aliquots of 225 μL of the sample were transferred onto three cold Brucella agar plates with 5% sheep blood and 5% fetal bovine serum and then incubated for 12 h at 37 °C under a humidified atmosphere of 5% CO_2_. After incubation, the cells were inoculated and streaked onto three cold Brucella agar plates with 5% sheep blood, 5% fetal bovine serum and 8 μg/mL of chloramphenicol (selective antibiotic). The plates were incubated for five days under microaerobic conditions as described above.

At the end of the five-day incubation, all colonies that grew on the selective medium were harvested and propagated individually on a fresh cold Brucella agar plate with 5% sheep blood, 5% fetal bovine serum and 8 μg/mL of chloramphenicol and then incubated for 48 h at 37 °C under the microaerobic conditions as described above.

Finally, to confirm that homologous recombination was carried out, the DNA of all those *H. pylori* 26695 clones that grew in the selective medium was purified and sequenced with the primer sets FUS32 R5′F/FUS32 R5′R, FUS32 RMF/FUS32 RMR and FUS32 R3′F/FUS32 3′R (Table 1, Appendix A).

All recombinant *H. pylori* 26695 mutant (HpΔ*vapD*) strains were stored in Brucella broth with 15% glycerol cryotubes at −80 °C.

### 2.5. Bacterial Growth Curves Comparison

To determine whether the *vapD* gene mutation in the *H. pylori* 26695 wt strain triggered any alteration in the behavior or growth rate of the strain, parallel growth curves were performed for both *H. pylori* 26695 wt and HpΔ*vapD* strains.

#### 2.5.1. Bacterial Culture and Inoculation of Growth Curve

*H. pylori* 26695 wt and HpΔ*vapD* were cultured in Brucella agar plates with 5% sheep blood and incubated for 48 h at 37 °C under microaerobic conditions. A small sample was taken from each culture to adjust 10 mL of Brucella broth (BB) to a density equivalent to a McFarland tube 1 (3 × 10^8^ CFU/mL). From these stock suspensions, 1 mL was inoculated into flasks with 9 mL BB with 5% SFB (by duplication) at a final concentration of 3 × 10^7^ CFU/mL per flask. The samples were incubated for five days at 37 °C under microaerobic conditions. The time interval between each sample collection was 24 h.

Growth curves of *H. pylori* 26695 wt and HpΔ*vapD* were evaluated under the conditions outlined above. A non-inoculated flask was also included as a control sample and was used as a negative control for growth and blank for the OD 620 nm measurements.

At each time point [12 h (t1), 24 h (t2), 36 h (t3), 48 h (t4), 60 h (t5), 72 h (t6), 84 h (t7), 96 h (t8) and 108 h (t9)] of the curve, two aliquots of each sample were taken to perform three different measurements. An aliquot of 1 mL was used to determine total proteins. The viability of *H. pylori* 26695 wt and HpΔ*vapD* was determined by counting Colony Forming Units (CFUs). The average colony count for each strain was determined based on three different assays: by inoculate, 100 μL of diluted aliquots (1:10; 1:100) on Brucella agar plates supplemented with 5% sheep blood and 5% fetal bovine serum. A non-inoculated flask was also plated as a negative control. The plates were incubated for five days at 37 °C under a humidified atmosphere of 8% CO_2_, and then the colonies were counted. The results for each interval were estimated as the average of three independent repetitions, including three replicates per independent assessment. Finally, in parallel to previously depicted assays, the growth of each sample was measured by means of the absorbance (OD) of the culture at 620 nm in a visible light spectrophotometer (Tecan Instruments, Männedorf, Switzerland).

#### 2.5.2. Total Protein Extraction

The pellet (*H. pylori*-AGS) was resuspended in 500 µL of RIPA buffer (NaCl 150 mM, Tris-HCl 50 mM pH 7.4, EDTA 2 mM, Nonidet-40 1% *v*/*v*, sodium deoxycholate 0.5% *w*/*v*, SDS 0.05% *w*/*v*) supplemented with protease inhibitor cocktail, EDTA-free and PMSF 0.2 mM, and incubated for 30 min on ice with gentle agitation using an end-over-end rotator. The lysate was centrifuged at 15,000× *g* for 20 min at 4 °C, before recovering the supernatant. The protein was quantified by the Bradford method at 595 nm.

### 2.6. H. pylori 26695 wt-AGS Cell and H. pylori 26695 ΔvapD-AGS Cell Co-Cultures

AGS cells (ATCC^®^ CRL-1739) were grown in Dulbecco’s Modified Eagle Medium (DMEM, Life Technologies^®^, Woodland, CA, USA), supplemented with 5% fetal bovine serum (FBS, Corning Costar, Manassas, VA, USA) and antibiotics. The AGS cells were incubated in 5% CO_2_ atmosphere until 80% confluency was reached and then distributed into 12-well plates. *H. pylori* 26695 wt and HpΔ*vapD* strains were grown in blood agar plates (BAPs) supplemented with 5% fetal bovine serum for 48 h in microaerobic conditions. AGS cells (3.2 × 10^5^) with fresh DMEM were inoculated separately with *H. pylori* 26695 wt strain and HpΔ*vapD* strain to a 3.2 × 10^7^ CFU/mL inoculum. All the resulting co-cultures were incubated, cultivated and worked as previously mentioned by Morales-Espinosa et al. [4] (Table 1).

Detection of transcription levels of *vacA, vapD*, *ureA* and 16s Hp genes from *H. pylori* 26695 wt and HpΔ*vapD* strains in the intracellular environment. Total RNA was isolated from the cellular package (intracellular *H. pylori*-AGS cells) at different time points using Hybrid-R^TM^ (GeneAll Biotechnology Co., Ltd, Song-gu, Seoul, Korea) according to the manufacturer’s instructions.

Total RNA (1 µg) was reverse transcribed using a QuantiTect^®^ Reverse Transcription Kit (QIAGEN^®^, Hilden, Germany) according to the manufacturer’s instructions. *vacA*, *vapD, 16s Hp, ureA* and GAPDH (life Technologies) mRNA levels were determined using real-time PCR and a step one plus Real-Time PCR system (Applied Biosystems^®^, Singapore) as Morales-Espinosa et al. mentioned previously [4].

## 3. Results

We determined the growth behavior of the *H. pylori* 26695 wt and HpΔ*vapD* strains, which were grown in vitro (synthetic media) in the laboratory at different time points (from 0 to 96 h). An aliquot of 1 mL was used to obtain CFU count, optical density (O.D) and total proteins. We observed that both strains of *H. pylori* 26695 (wt and HpΔ*vapD*) showed the same behavior, suggesting that HpΔ*vapD* did not affect the growth rate of *H. pylori* in synthetic media. This suggests that the *vapD* gene is not necessary when the bacterium is growing outside of the eukaryote cells (Figure 2).

We also wanted to know if the *vapD* gene mutation would affect the survival of *H. pylori* inside gastric cells, so we performed parallel assays of both *H. pylori* 26695 strains (wt and HpΔ*vapD*) in co-culture with AGS cells, which were maintained up to 108 h (t9). A CFU count from each co-culture was determined from the intracellular niche.

In terms of growth rate, our results showed a significant difference (*p* = 0.002) between the two strains. While the number of CFUs from the *H. pylori* 26695 wt strain was maintained throughout the assay (12–108 h), the number of HpΔ*vapD* strain colonies identified from co-culture with AGS cells decreased significantly after 36 h until it disappeared completely (Figure 3).

However, when the intracellular transcription levels of the *vacA* gene from *H. pylori* 26695 wt was compared with the HpΔ*vapD* strain (taking the transcription level of the endogenous bacterial gene 16s as a reference value), we observed that the *vacA* gene from *H. pylori* 26695 wt had very high transcription levels in comparison to *vacA* from the HpΔ*vapD* strain (Figure 4, Appendix A). In the *H. pylori* wt strain, *vacA* was transcribed up to 10,000-fold higher than *vacA* from the HpΔ*vapD* strain, with a significant difference of *p* = 0.005 (Figure 4, Appendix A). This suggests metabolic impairment of *vapD*-mutated *H. pylori* within the intracellular environment.

With regard to the *ure*A gene, we could not determine transcription levels of this gene in both *H. pylori* 26695 (wt and Δ*vapD* mutant) strains in the intracellular environment.

## 4. Discussion

In a previous study, we showed that *vapD* is a strain-specific gene, and approximately 38% of the Mexican strains of *H. pylori* present this gene [14], although its frequency varies depending on the geographic region and the population type [13]. Initially, this gene was described by Cao et al. in a hypervariable chromosomal region of the *H. pylori* 60190 strain and in other microorganisms, such as *Rhodococcus equi*, non-typeable *Haemophilus influenzae*, *Dichelobacter nodosus*, *Neisseriae gonorrhoea* and other microorganisms of different phyla and genera, where it was shown that the *vapD* gene had been acquired by horizontal gene transfer [15]. However, in *H. pylori*, *vapD* became fixed in its chromosome once it was acquired. In *Rhodococcus equi*, the *vapD* gene is highly inducible inside macrophages and participates in preventing the fusion of the phagolysosome [7]. In non-typeable *Haemophilus influenzae* (NTHi), the *vapD* gene forms part of the toxin–antitoxin module (*vapXD*), which is involved in metabolic regulation of bacteria by enabling a switch to a dormant state under stress conditions. Ren et al. (2012) concluded that the *vapXD* TA locus enhances NTHi survival and virulence during infection in vitro and in vivo using a mechanism of mRNA cleavage [8]. Mutations of the *vapD* gene have shown reduced NTHi persistence in the chinchilla model of otitis media, leading to the belief that *vapD* is involved in the persistence of NTHi within otic epithelial cells [8].

In *H. pylori*, the role of *vapD* is unknown, although a relationship between the Cas2 family of ribonuclease associated with the CRISPR system of microbial immunity and VapD has been suggested [5]. Contrary to what was reported by Chakraborty and Chatterjee [16], the *vapD* gene of the HP0315 locus in the *H. pylori* 26695 strain is not a component of the TA system, and no genes associated with it as antitoxins are found, as corroborated by Gilep [17] and Kwon [5], which suggests that it may be an evolutionary intermediate that does not belong to either the Cas2 family or the TA system. There have been very few studies carried out on the *vapD* gene and its protein. In previous studies carried out by our group, we tested the *vapD*-positive *H. pylori* 26695 strain in co-culture with AGS cells, where *H. pylori* remained metabolically active inside AGS cells up to 108 h (time chosen by our experiment). In that study, the *vapD* transcription levels were measured by RT-PCR, and the assay showed high transcription levels of the *vapD* gene, reaching a maximum level of transcription after 96 h of being inside the AGS cells [4]. In this same study, it was also demonstrated that the transcription of the *vapD* gene occurred in gastric biopsies (in vivo) from patients with severe gastric pathologies, such as chronic atrophy gastritis, follicular gastritis, peptic ulcer, gastric metaplasia and gastric cancer, and the results showed high levels of *vapD* transcription from the gastric antrum and corpus in all patients, indicating that *vapD*-positive *H. pylori* strains were colonizing both anatomic sites at the same time that the *vapD* gene was being transcribed in vivo [4]. In another study, our group produced polyclonal antibodies against the recombinant VapD protein of *H. pylori* [18], and they were used in immunofluorescence assays to visualize the VapD protein of *H. pylori* in the cytoplasm of AGS cells in co-cultures at different times (manuscript in preparation). It is well known that different proteins or virulence factors of *H. pylori* are expressed into gastric cells and under different environmental conditions [4,19,20,21].

The relevance and importance of the invasiveness of any microorganism could be an advantage in its pathogenicity, favoring its permanence, multiplication and dissemination into other cells, since within the cells it is protective against both innate and acquired immune systems and antibiotics, as well as having an important source of nutrients [22]. However, the microorganisms need strategies that allow survival within eukaryotic cells, such as taking refuge within a vesicular structure, avoiding intracellular antimicrobial defenses, inhibiting the fusion of phagolysosomes or preventing apoptosis, ensuring, in turn, that the cell maintains its division cycle [23,24]. Actually, it is well documented that there are invasive strains of *H. pylori* [25,26,27] that enter gastric cells via a zipper-like phagocytic mechanism that depends on protein kinase C and phosphatidylinositol 3-kinase [28]. Once inside, they find refuge within large vacuolar compartments in which *H. pylori* can persist for a long time [25,26,27]. The VacA protein is an important virulence factor and has been identified as being responsible for the formation of vacuoles mediated by late endosomal compartments that depend on vacuolar ATPase, the small GTPase Rab7, dynamin and syntaxin 7 [29]. The vacuolar phenotype of the strains results in the formation and maintenance of an intracellular niche for *H. pylori* [29], which may contribute to the protection of bacteria inside AGS cells. However, this does not provide a satisfactory explanation of the molecular mechanism responsible for the intracellular survival of *H. pylori* inside gastric epithelial cells.

In the current study, we wanted to determine if the mutation of the *vapD* gene in the *H. pylori* 26695 strain could affect its persistence and survival within the gastric epithelial cell. With this in mind, we performed parallel assays of co-cultures of AGS cells with *H. pylori* 26695 wt and HpΔ*vapD* strains. Our results show that the *H. pylori* 26695 wt strain behaved within AGS cells in a similar way to that previously reported, in which *H. pylori* invades AGS cells in the first 6 h of co-culture: the VapD expression increased while it remained for a longer time in the intracellular niche [4], and CFUs were recovered in sufficient amounts at different time points. Meanwhile, co-infection of AGS cells by the HpΔ*vapD* strain also maintained its ability to invade AGS cells in the first 6 h, but after that, its growth curve was significantly affected, which was reflected in the gradual decrease in the number of CFUs in AGS cells at different times points until none were found. These results were corroborated by the significant decrease (*p* = 0.0005) in the expression levels of VacA of this HpΔ*vapD* strain (in relation to the *H. pylori* 26695 wt strain), until they became imperceptible, probably due to an important decrease in the number of *H. pylori* within the cell or due to the death of the bacteria in the intracellular environment, suggesting that the *vapD* gene is somehow involved in the survival of *H. pylori* within gastric cells.

Contrary to what was observed in the intracellular environment, the growth of both strains in vitro (synthetic medium) shows no difference between the development (bacterial optical density curve, bacterial growth curve and protein concentration) of both strains, indicating that the *vapD* gene is silenced and that it is not necessary when the bacterium is growing outside eukaryote cells. Our results suggest that the VapD protein is participating in the persistence of *H. pylori* within gastric cells.

Further studies will have to be carried out to detail the mechanism of action of *vapD* and its participation in the survival of the bacteria within gastric cells.

## 5. Conclusions

The results obtained in the current study suggest that *vapD* plays a role in the survival and persistence of *H. pylori* in the intracellular environment, and this needs further study to determine its precise role.

Our results suggest that this intracellular characteristic of some *H. pylori* strains increases the risk of developing chronic infection, contributes to treatment failure and helps the chronic inflammatory process that promotes the development of gastric cancer.

## Figures and Tables

**Figure 1 microorganisms-13-01952-f001:**
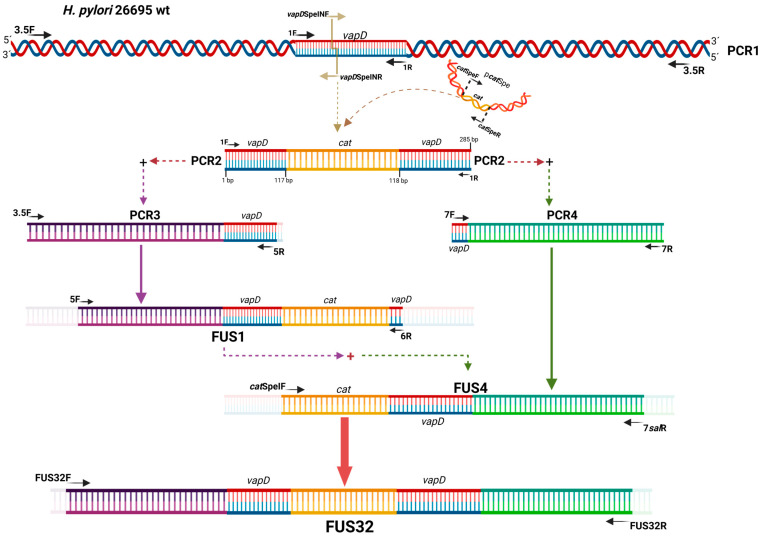
*vapD* gene knockout construction. The *vapD* gene knockout construction was performed by *cat* cassette insertion between nucleotides 117 and 118 of *vapD* ORF (HP315) of the *H. pylori* 26695 strain.

**Figure 2 microorganisms-13-01952-f002:**
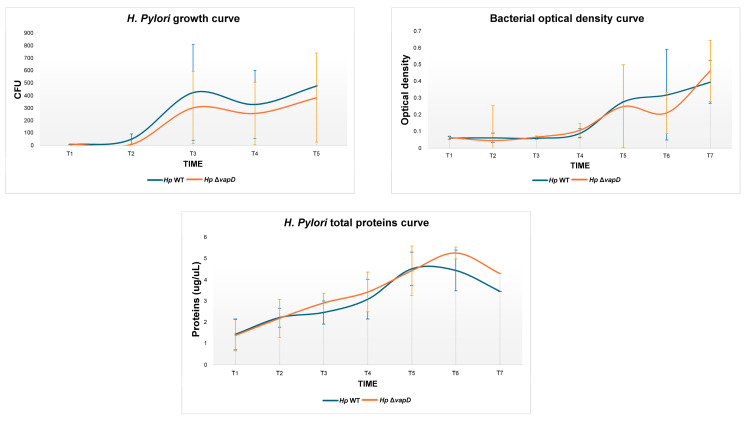
Growth behavior of *H. pylori* 26695 wild-type (wt) strain and *H. pylori* 26695 Δ*vapD* strain under in vitro conditions (synthetic media) at different time points. The figure depicts the growth curve of the *H. pylori* wt strain (blue line) and *H. pylori* 26695Δ*vapD* strain (red line) obtained by CFU count, optical density, and total protein curve.

**Figure 3 microorganisms-13-01952-f003:**
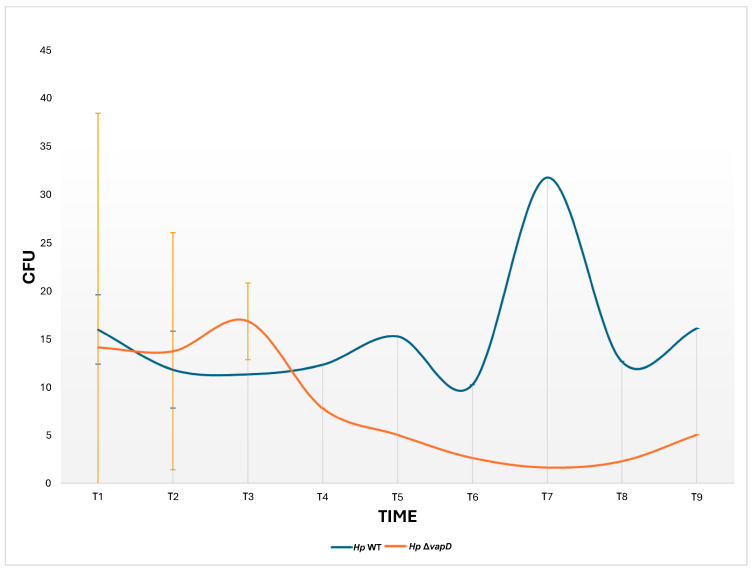
Growth curves of the *H. pylori* 26695 wild-type (wt) strain and *H. pylori* 26695Δ*vapD* strain from the intracellular environment at different time points. This shows the growth curve of the *H. pylori* wt strain (blue line) and *H. pylori* 26695 Δ*vapD* mutant strain (red line) in co-culture with AGS cells, which was determined by CFU count obtained from the intracellular environment. The results show a significant difference (*p* = 0.002) between the two strains.

**Figure 4 microorganisms-13-01952-f004:**
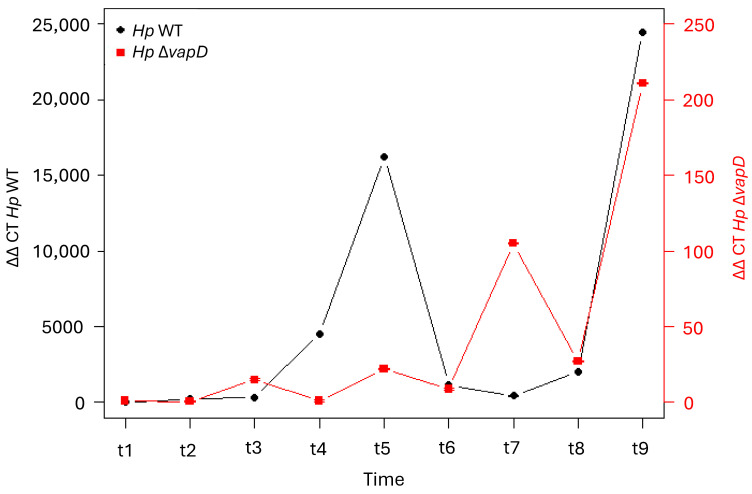
Relation of transcription level of *vacA* with the 16s endogenous control gene of *H. pylori* 26695 wild-type strain and *H. pylori* 26695 Δ*vapD* strain. *H. pylori* wt strain (black line) and *H. pylori* 26695 Δ*vapD* mutant strain (red line). A significant difference in the *vacA* transcription levels can be observed between both strains (*p* = 0.005).

**Table 1 microorganisms-13-01952-t001:** Primers used in the construction of *vapD* knockouts and in the detection of different transcription levels of genes by RT-PCR.

Primers	Forward	Reverse	MW	Reference
1 F/R	5′ATGTATGCTTTAGCGTTTG 3′	5′GGATTTCACAATCTCAGTAA 3′	285 bp	[13]
*cat*SpeI F/R	5′GTTGATCGACTAGTAAGAGGTTC 3′	5′GCCATTCAACTAGTTATTATCACT 3′	900 bp	This study
*vapD*SpeIN F/R	5′TGACTAGTCTCAAGGGAG 3′	5′GAGACTAGTCAAACCCTAATAG 3′		This study
3.5 F/R	5′AAACGCGCAAAATCAAAACAACTT 3′	5′CGCGCAAGAAATGAGCAATAA 3′	3.5 Kb	This study
5 F/R	5′ATCGCTCACTTTGGCACTCA 3′	5′TAGGCTTTATTGTAGGGTTCTCCG 3′	1.9 Kb	This study
6 F/R	5′ACGCGCAAAATCAAAACAA 3′	5′TAAACGCTCTAATATCCCTAACAG 3′	3.2 Kb	This study
7 F/R	5′GACTTTAGCGATTTTACTGAGATT 3′	5′GCCATTTAGAGCGTGAA 3′	1.7 Kb	This study
7 *sal*R		5′GGATTTTCGTCGACATCAAGGGTT 3′		This study
PCR3 (3.5F/5R)	5′AAACGCGCAAAATCAAAACAACTT 3′	5′TAGGCTTTATTGTAGGGTTCTCCG 3′	2.2 Kb	This study
PCR4 (7F/7R)	5′GACTTTAGCGATTTTACTGAGATT 3′	5′GACTTTAGCGATTTTACTGAGATT 3′	1.7 Kb	This study
FUS1 (5F/6R)	5′ATCGCTCACTTTGGCACTCA 3′	5′TAAACGCTCTAATATCCCTAACAG 3′	2.9 Kb	This study
FUS4 (*cat*SpeIF/7*sal*R)	5′GTTGATCGACTAGTAAGAGGTTC 3′	5′GGATTTTCGTCGACATCAAGGGTT 3′	2.6 Kb	This study
FUS32 F/R	5′ACCACCGCGCTCTCCAAAGTC 3′	5′AACGCCAGATCCAAAGCCAAAAGA 3′	3.6 Kb	This study
FUS32 R5′F/R	5′ACACAAAATACAGCGAAAAACAGC 3′	5′ATCAGGCGGGCAAGAATG 3′	754 bp	This study
FUS32 RM F/R	5′AGAATACGGAGAACCCTACAATAA 3′	5′CCAGCGGCATCAGCACCTT 3′	727 bp	This study
FUS32 R3′F/R	5′CAAGGCGACAAGGTGCTGATGC 3′	5′CCCCACGATTGAATGAAAAAGAGT 3′	714 bp	This study
qPCR Primers	Forward	Reverse		
*ureA* F/R	5′AGTTCCTGGTGAGTTGTTCTT 3′	5′TGGAAGTGTGAGCCGATTT 3′	120 bp	This study
*vacA* F/R	5′ATGGAAATACAACAAACACACC 3′	5′CCAACAATGGCTGGAATGA 3′	137 bp	This study
*vapD* F/R	5′ATGTATGCTTTAGCGTTTG 3′	5′GGATTTCACAATCTCAGTAA 3′	285 bp	[13]
16sHp F/R	5′GCAAGCGTTACTCGGAATCA 3′	5′ACCTACCTCTCCCACACTCTA 3′	126 bp	This study
TaqMan probes	MGB Probe			
*ureA*	NED-TGAAGACATCACTATCAACGAAGGCA		26 bp	This study
** *vacA* **	VIC-ACTTTGTTGCGGTGTGATGCTGAC		24 bp	This study
*vapD*	FAM-AGAGCGTTTAAGGTAGAGGACTTTAGCGA		29 bp	This study
16sHp	NED-TAGGCGGGATAGTCAGTCAGGTGT		24 bp	This study

## Data Availability

The original contributions presented in this study are included in the article/supplementary material. Further inquiries can be directed to the corresponding author.

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
