# Peer review of "vapD Mutation Shows Impairment in the Persistence of Helicobacter pylori Within AGS Cells"

_microorganisms, 2025, doi:10.3390/microorganisms13081952_

Round 1

Reviewer 1 Report

Comments and Suggestions for Authors

The study investigates the role of vapD in Helicobacter pylori survival within gastric cells using a knockout strain (Hp ΔvapD). The experimental design is innovative and methodologically sound, particularly in demonstrating that vapD is dispensable for extracellular growth but essential for intracellular maintenance in AGS cells. The dramatic reduction in vacA transcription in the mutant strain is a striking finding, suggesting a potential regulatory link between vapD and key virulence factors. These results advance understanding of H. pylori pathogenesis and highlight vapD as a critical intracellular persistence factor.

However, 1.The rationale for targeting vapD requires expansion. vapD (virulence-associated protein D) is conserved across bacteria (e.g., MycoplasmaCampylobacter) and linked to stress response, plasmid stability, and host adaptation et. al.Omitting this background diminishes the significance of your findings. 2.The cited references are too old. It is recommended to update them and cite some of the latest literature on the intracellular transcriptomics and proteomics of VapD.

Comments on the Quality of English Language

The professionalism and quality of the English writing are acceptable.

Author Response

We deeply appreciate your review of this manuscript and considering your suggestions, we have expanded the introductory framework in relation to VapD and its importance among other microorganisms. In the references, we cite the most recently published documents related to research conducted with the VapD protein.

Reviewer 2 Report

Comments and Suggestions for Authors

Dear Authors,

This manuscript presents a well-structured investigation into the role of the vapD gene in the intracellular persistence of Helicobacter pylori within human gastric adenocarcinoma (AGS) cells. The Authors create a targeted vapD knockout in the H. pylori 26695 strain (Hp ΔvapD) and assess its phenotype in both synthetic media and intracellular environments (AGS cells). Moreover, the Authors describe in details bacterial and cell culture conditions, molecular cloning steps for vapD and chloramphenicol resistance cassette (Cmr), the design and execution of overlap extension PCR to insert homologous flanking regions, and others.

However, some minor clarifications, and editorial adjustments are recommended to improve the clarity, logical connection, and readability of the manuscript.

Major Strengths:

  1. The authors employ a robust combination of molecular biology and microbiological approaches, including CFU enumeration, total protein quantification, and qRT-PCR to compare gene expression between wild-type and mutant strains. These methods are appropriate for evaluating both bacterial viability and transcriptional responses in the host cell environment.
  2. The observation that the vapD mutation does not affect growth in synthetic media but significantly impairs bacterial recovery from AGS cells after 36 hours is compelling. It indicates a specific role for vapD in intracellular persistence, rather than general viability.
  3. The dramatic (>10,000-fold) downregulation of vacA transcripts in the Hp ΔvapD strain suggests a potential regulatory or functional relationship between vapD and virulence factor expression. This finding is highly relevant to understanding H. pylori's capacity to modulate host-pathogen interactions during intracellular residency.

Suggestions for consideration:

  1. Please consider follow-up experiments such as transcriptomics (e.g., RNA-seq), its could elucidate the regulatory network involved.
  2. Consider the assessment the effect of vapD mutation on host cell responses (e.g., apoptosis, cytokine production…), it could offer additional insights into how vapD contributes to H. pylori's intracellular survival strategy.

Moreover:

  1. PCR confirmation of insert presence is noted; consider sequencing, which confirmed the insert identity would strengthen this section.
  2. Including the vector map or schematic of constructs such as pcatSpe and pvapD in supplementary materials would be helpful.
  3. Repeating the specific overlap PCR conditions multiple times introduces redundancy. Consider condensing and referring to a supplemental methods section.
  4. Several sections refer to “vapD gen” instead of “vapD gene.” Please revise throughout.
  5. Terms such as “spin down the cell” should be revised for clarity, e.g., “centrifuged the cells.”
  6. Ensure uniform formatting of primer sequences (e.g., consistent use of 5′ and 3′ symbols, line breaks.
  7. Please consider writing pylori in italic throughout the text.
  8. I suggest changing µg/ml to µg/mL throughout the text.
  9. Section 2.5.2: is “4ºC” and should be “4℃”.
  10. Units such as “900 pb” should be corrected to “900 bp”.
  11. The degree symbol “0C” is inconsistently used—should be corrected to “°C”.
  12. Consider improvement in the entire text “3X105” on “3x105”.
  13. Figure 2: is “synthetic medio” and should be “… media”.
  14. Figure 4: in the description of the figure it is written “ pylori wt strain (blue line)” and in the figure it is black.
  15. Penultimate line on page 12: please consider writing "our results suggest that ...
  16. Please also consider presenting a figure that depicts the total protein curves for wt and mutant in the intercellular environment.
  17. Try to use passive voice consistently throughout the text. It is often preferred in results sections.

I recommend the article for publication in Microorganisms after minor revision.

Author Response

Suggestions for consideration:

  1. Please consider follow-up experiments such as transcriptomics (e.g., RNA-seq), it could elucidate the regulatory network involved.

Answer: We are deeply grateful for the reviewer comments and suggestions, which reaffirms the suitability of our experimental design, as we are currently waiting for the first RNA- seqs from nine cultures of the H. pylori 26695 wt and HpΔvapD strains. Furthermore, following this line of thought, we are standardizing the methods necessary to continue with proteomics.

  1. Consider the assessment the effect of vapD mutation on host cell responses (e.g., apoptosis, cytokine production…), it could offer additional insights into how vapD contributes to pylori's intracellular survival strategy.

Answer: We sincerely thank the reviewer once again for the time and effort dedicated to evaluating our work, the constructive feedback and perspective provided. Of course, we will consider in the short-term conducting the necessary assays to evaluate the different cellular responses of both the host and H. pylori.

Moreover:

  1. PCR confirmation of insert presence is noted; consider sequencing, which confirmed the insert identity would strengthen this section.

Answer: Done. We add in a supplementary figure the sequencing of the insert (Figure S2).

  1. Including the vector map or schematic of constructs such as pcatSpe and pvapD in supplementary materials would be helpful.

Answer: In supplementary material, we include the Figures S1A and S1B, which illustrate the constructions of the vectors pcatSpe and pvapD.

  1. Repeating the specific overlap PCR conditions multiple times introduces redundancy. Consider condensing and referring to a supplemental methods section.

Answer: done (Table S1)

  1. Several sections refer to “vapD gen” instead of “vapD gene. Please revise throughout.

Answer: It was corrected throughout the paper

  1. Terms such as “spin down the cell” should be revised for clarity, e.g., “centrifuged the cells.”

Answer: It was corrected

  1. Ensure uniform formatting of primer sequences (e.g., consistent use of 5′ and 3′ symbols, line breaks.

Answer: Done

  1. Please consider writing pylori in italic throughout the text.

Answer: Done

  1. I suggest changing μg/ml to μg/mL throughout the text.

Answer: Done

  1. Section 2.5.2: is “4ºC” and should be “4℃”.

Answer: Done

  1. Units such as “900 pb” should be corrected to “900 bp”.

Answer: pb was changed to bp throughout all the paper.

  1. The degree symbol “0C”” is inconsistently used—should be

corrected to “°C”

Answer: The degree symbol was corrected and homogenized throughout the article.

  1. Consider improvement in the entire text “3X105” on “3x105”.

Answer: Done. It was improved throughout the text, as suggested by the reviewer.

  1. Figure 2: is “synthetic medio” and should be “… media”.

Answer: was changed to “synthetic media”

  1. Figure 4: in the description of the figure, it is written “pylori wt strain (blue line)” and in the figure it is black.

Answer: was changed to “black line”

  1. Penultimate line on page 12: please consider writing "our results suggest that ...

Answer: Done

  1. Please also consider presenting a figure that depicts the total protein curves for wt and mutant in the intercellular environment.

Answer: Dear reviewer, the problem with these curves (despite several assays) was that we did not obtain precise and reproducible values for the specific proteins of intracellular H. pylori. I mean, we always had (greater or lesser extent) a background of proteins from the AGS cells. This was mainly since we did not find any technique that could eliminate the proteins and debris from the AGS cells after treating them with saponin to break them down and release the intracellular bacteria. This was why we did not include these curves.